# CORD-19: The COVID-19 Open Research Dataset

**Lucy Lu Wang**[1,*]    **Kyle Lo**[1,*]    **Yoganand Chandrasekhar**[1]    **Russell Reas**[1]
**Jiangjiang Yang**[1]    **Douglas Burdick**[2]    **Darrin Eide**[3]    **Kathryn Funk**[4]
**Yannis Katsis**[2]    **Rodney Kinney**[1]    **Yunyao Li**[2]    **Ziyang Liu**[6]
**William Merrill**[1]    **Paul Mooney**[5]    **Dewey Murdick**[7]    **Devvret Rishi**[5]
**Jerry Sheehan**[4]    **Zhihong Shen**[3]    **Brandon Stilson**[1]    **Alex D. Wade**[6]
**Kuansan Wang**[3]    **Nancy Xin Ru Wang**[2]    **Chris Wilhelm**[1]    **Boya Xie**[3]
**Douglas Raymond**[1]    **Daniel S. Weld**[1,8]    **Oren Etzioni**[1]    **Sebastian Kohlmeier**[1]

[1]Allen Institute for AI    [2] IBM Research    [3]Microsoft Research
[4]National Library of Medicine    [5]Kaggle    [6]Chan Zuckerberg Initiative
[7]Georgetown University    [8]University of Washington
{lucyw, kylel}@allenai.org

## Abstract

The COVID-19 Open Research Dataset (CORD-19) is a growing[1] resource of scientific papers on COVID-19 and related historical coronavirus research. CORD-19 is designed to facilitate the development of text mining and information retrieval systems over its rich collection of metadata and structured full text papers. Since its release, CORD-19 has been downloaded[2] over 200K times and has served as the basis of many COVID-19 text mining and discovery systems. In this article, we describe the mechanics of dataset construction, highlighting challenges and key design decisions, provide an overview of how CORD-19 has been used, and describe several shared tasks built around the dataset. We hope this resource will continue to bring together the computing community, biomedical experts, and policy makers in the search for effective treatments and management policies for COVID-19.

## 1 Introduction

On March 16, 2020, the Allen Institute for AI (AI2), in collaboration with our partners at The White House Office of Science and Technology Policy (OSTP), the National Library of Medicine (NLM), the Chan Zuckerburg Initiative (CZI), Microsoft Research, and Kaggle, coordinated by Georgetown University's Center for Security and Emerging Technology (CSET), released the first version

---

*denotes equal contribution

[1]The dataset continues to be updated daily with papers from new sources and the latest publications. Statistics reported in this article are up-to-date as of version 2020-06-14.

[2]https://www.semanticscholar.org/cord19

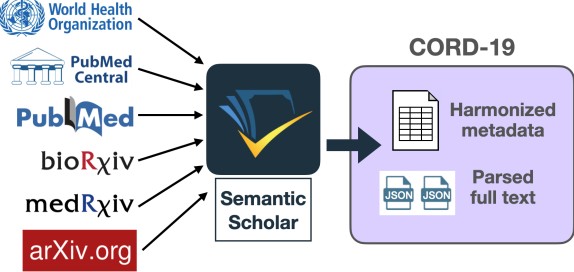

Figure 1: Papers and preprints are collected from different sources through Semantic Scholar. Released as part of CORD-19 are the harmonized and deduplicated metadata and full text JSON.

of CORD-19. This resource is a large and growing collection of publications and preprints on COVID-19 and related historical coronaviruses such as SARS and MERS. The initial release consisted of 28K papers, and the collection has grown to more than 140K papers over the subsequent weeks. Papers and preprints from several archives are collected and ingested through the Semantic Scholar literature search engine,[3] metadata are harmonized and deduplicated, and paper documents are processed through the pipeline established in Lo et al. (2020) to extract full text (more than 50% of papers in CORD-19 have full text). We commit to providing regular updates to the dataset until an end to the COVID-19 crisis is foreseeable.

CORD-19 aims to connect the machine learning community with biomedical domain experts and policy makers in the race to identify effective treatments and management policies for COVID-19. The goal is to harness these diverse and com-

---

[3]https://semanticscholar.org/

plementary pools of expertise to discover relevant information more quickly from the literature. Users of the dataset have leveraged AI-based techniques in information retrieval and natural language processing to extract useful information.

Responses to CORD-19 have been overwhelmingly positive, with the dataset being downloaded over 200K times in the three months since its release. The dataset has been used by clinicians and clinical researchers to conduct systematic reviews, has been leveraged by data scientists and machine learning practitioners to construct search and extraction tools, and is being used as the foundation for several successful shared tasks. We summarize research and shared tasks in Section 4.

In this article, we briefly describe:

1. The content and creation of CORD-19,
2. Design decisions and challenges around creating the dataset,
3. Research conducted on the dataset, and how shared tasks have facilitated this research, and
4. A roadmap for CORD-19 going forward.

## 2 Dataset

CORD-19 integrates papers and preprints from several sources (Figure 1), where a paper is defined as the base unit of published knowledge, and a preprint as an unpublished but publicly available counterpart of a paper. Throughout the rest of Section 2, we discuss papers, though the same processing steps are adopted for preprints. First, we ingest into Semantic Scholar paper metadata and documents from each source. Each paper is associated with bibliographic metadata, like title, authors, publication venue, etc, as well as unique identifiers such as a DOI, PubMed Central ID, PubMed ID, the WHO Covidence #,[4] MAG identifier (Shen et al., 2018), and others. Some papers are associated with documents, the physical artifacts containing paper content; these are the familiar PDFs, XMLs, or physical print-outs we read.

For the CORD-19 effort, we generate harmonized and deduplicated metadata as well as structured full text parses of paper documents as output. We provide full text parses in cases where we have access to the paper documents, and where the documents are available under an open access license

---

[4]https://www.who.int/emergencies/diseases/novel-coronavirus-2019/global-research-on-novel-coronavirus-2019-ncov

(e.g. Creative Commons (CC),[5] publisher-specific COVID-19 licenses,[6] or identified as open access through DOI lookup in the Unpaywall[7] database).

### 2.1 Sources of papers

Papers in CORD-19 are sourced from PubMed Central (PMC), PubMed, the World Health Organization's Covid-19 Database,[4] and preprint servers bioRxiv, medRxiv, and arXiv. The PMC Public Health Emergency Covid-19 Initiative[6] expanded access to COVID-19 literature by working with publishers to make coronavirus-related papers discoverable and accessible through PMC under open access license terms that allow for reuse and secondary analysis. BioRxiv and medRxiv preprints were initially provided by CZI, and are now ingested through Semantic Scholar along with all other included sources. We also work directly with publishers such as Elsevier[8] and Springer Nature,[9] to provide full text coverage of relevant papers available in their back catalog.

All papers are retrieved given the query[10]:

```
"COVID" OR "COVID-19" OR
"Coronavirus" OR "Corona virus"
OR "2019-nCoV" OR "SARS-CoV"
OR "MERS-CoV" OR "Severe Acute
Respiratory Syndrome" OR "Middle
East Respiratory Syndrome"
```

Papers that match on these keywords in their title, abstract, or body text are included in the dataset. Query expansion is performed by PMC on these search terms, affecting the subset of papers in CORD-19 retrieved from PMC.

### 2.2 Processing metadata

The initial collection of sourced papers suffers from duplication and incomplete or conflicting metadata. We perform the following operations to harmonize and deduplicate all metadata:

1. Cluster papers using paper identifiers
2. Select canonical metadata for each cluster
3. Filter clusters to remove unwanted entries

---

[5]https://creativecommons.org/
[6]https://www.ncbi.nlm.nih.gov/pmc/about/covid-19/
[7]https://unpaywall.org/
[8]https://www.elsevier.com/connect/coronavirus-information-center
[9]https://www.springernature.com/gp/researchers/campaigns/coronavirus
[10]Adapted from the Elsevier COVID-19 site[8]

**Clustering papers** We cluster papers if they overlap on any of the following identifiers: {*doi, pmc_id, pubmed_id, arxiv_id, who_covidence_id, mag_id*}. If two papers from different sources have an identifier in common and no other identifier conflicts between them, we assign them to the same cluster. Each cluster is assigned a unique identifier **CORD_UID**, which persists between dataset releases. No existing identifier, such as DOI or PMC ID, is sufficient as the primary CORD-19 identifier. Some papers in PMC do not have DOIs; some papers from the WHO, publishers, or preprint servers like arXiv do not have PMC IDs or DOIs.

Occasionally, conflicts occur. For example, a paper $c$ with $(doi, pmc\_id, pubmed\_id)$ identifiers $(x, null, z')$ might share identifier $x$ with a cluster of papers $\{a, b\}$ that has identifiers $(x, y, z)$, but has a conflict $z' \neq z$. In this case, we choose to create a new cluster $\{c\}$, containing only paper $c$.[11]

**Selecting canonical metadata** Among each cluster, the canonical entry is selected to prioritize the availability of document files and the most permissive license. For example, between two papers with PDFs, one available under a CC license and one under a more restrictive COVID-19-specific copyright license, we select the CC-licensed paper entry as canonical. If any metadata in the canonical entry are missing, values from other members of the cluster are promoted to fill in the blanks.

**Cluster filtering** Some entries harvested from sources are not papers, and instead correspond to materials like tables of contents, indices, or informational documents. These entries are identified in an ad hoc manner and removed from the dataset.

### 2.3 Processing full text

Most papers are associated with one or more PDFs.[12] To extract full text and bibliographies from each PDF, we use the PDF parsing pipeline created for the S2ORC dataset (Lo et al., 2020).[13] In (Lo et al., 2020), we introduce the S2ORC JSON format for representing scientific paper full text,

which is used as the target output for paper full text in CORD-19. The pipeline involves:

1. Parse all PDFs to TEI XML files using GROBID[15] (Lopez, 2009)
2. Parse all TEI XML files to S2ORC JSON
3. Postprocess to clean up links between inline citations and bibliography entries.

We additionally parse JATS XML[16] files available for PMC papers using a custom parser, generating the same target S2ORC JSON format.

This creates two sets of full text JSON parses associated with the papers in the collection, one set originating from PDFs (available from more sources), and one set originating from JATS XML (available only for PMC papers). Each PDF parse has an associated SHA, the 40-digit SHA-1 of the associated PDF file, while each XML parse is named using its associated PMC ID. Around 48% of CORD-19 papers have an associated PDF parse, and around 37% have an XML parse, with the latter nearly a subset of the former. Most PDFs (>90%) are successfully parsed. Around 2.6% of CORD-19 papers are associated with multiple PDF SHA, due to a combination of paper clustering and the existence of supplementary PDF files.

### 2.4 Table parsing

Since the May 12, 2020 release of CORD-19, we also release selected HTML table parses. Tables contain important numeric and descriptive information such as sample sizes and results, which are the targets of many information extraction systems. A separate PDF table processing pipeline is used, consisting of table extraction and table understanding. *Table extraction* is based on the Smart Document Understanding (SDU) capability included in IBM Watson Discovery.[17] SDU converts a given PDF document from its native binary representation into a text-based representation like HTML which includes both identified document structures (e.g., tables, section headings, lists) and formatting information (e.g. positions for extracted text). *Table understanding* (also part of Watson Discovery) then annotates the extracted tables with additional semantic information, such as column and row headers and table captions. We leverage the Global Table Extractor (GTE) (Zheng et al.,

---

[11]This is a conservative clustering policy in which any metadata conflict prohibits clustering. An alternative policy would be to cluster if any identifier matches, under which $a$, $b$, and $c$ would form one cluster with identifiers $(x, y, [z, z'])$.

[12]PMC papers can have multiple associated PDFs per paper, separating the main text from supplementary materials.

[13]One major difference in full text parsing for CORD-19 is that we do not use ScienceParse,[14] as we always derive this metadata from the sources directly.

[14]https://github.com/allenai/science-parse

[15]https://github.com/kermitt2/grobid

[16]https://jats.nlm.nih.gov/

[17]https://www.ibm.com/cloud/watson-discovery

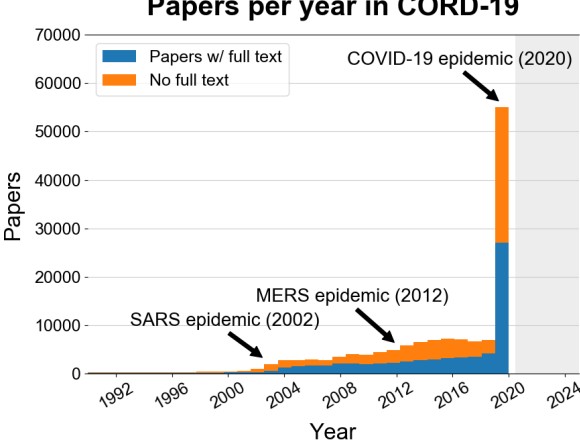

**Papers per year in CORD-19**

Figure 2: The distribution of papers per year in CORD-19. A spike in publications occurs in 2020 in response to COVID-19.

2020), which uses a specialized object detection and clustering technique to extract table bounding boxes and structures.

All PDFs are processed through this table extraction and understanding pipeline. If the Jaccard similarity of the table captions from the table parses and CORD-19 parses is above 0.9, we insert the HTML of the matched table into the full text JSON. We extract 188K tables from 54K documents, of which 33K tables are successfully matched to tables in 19K (around 25%) full text documents in CORD-19. Based on preliminary error analysis, we find that match failures are primarily due to caption mismatches between the two parse schemes. Thus, we plan to explore alternate matching functions, potentially leveraging table content and document location as additional features. See Appendix A for example table parses.

### 2.5 Dataset contents

CORD-19 has grown rapidly, now consisting of over 140K papers with over 72K full texts. Over 47K papers and 7K preprints on COVID-19 and coronaviruses have been released since the start of 2020, comprising nearly 40% of papers in the dataset.

Classification of CORD-19 papers to Microsoft Academic Graph (MAG) (Wang et al., 2019, 2020) fields of study (Shen et al., 2018) indicate that the dataset consists predominantly of papers in Medicine (55%), Biology (31%), and Chemistry (3%), which together constitute almost 90% of the corpus.[18] A breakdown of the most common MAG

---

[18]MAG identifier mappings are provided as a supplement

| Subfield | Count | % of corpus |
|---|---|---|
| Virology | 29567 | 25.5% |
| Immunology | 15954 | 13.8% |
| Surgery | 15667 | 13.5% |
| Internal medicine | 12045 | 10.4% |
| Intensive care medicine | 10624 | 9.2% |
| Molecular biology | 7268 | 6.3% |
| Pathology | 6611 | 5.7% |
| Genetics | 5231 | 4.5% |
| Other | 12997 | 11.2% |

Table 1: MAG subfield of study for CORD-19 papers.

subfields (L1 fields of study) represented in CORD-19 is given in Table 1.

Figure 2 shows the distribution of CORD-19 papers by date of publication. Coronavirus publications increased during and following the SARS and MERS epidemics, but the number of papers published in the early months of 2020 exploded in response to the COVID-19 epidemic. Using author affiliations in MAG, we identify the countries from which the research in CORD-19 is conducted. Large proportions of CORD-19 papers are associated with institutions based in the Americas (around 48K papers), Europe (over 35K papers), and Asia (over 30K papers).

## 3 Design decision & challenges

A number of challenges come into play in the creation of CORD-19. We summarize the primary design requirements of the dataset, along with challenges implicit within each requirement:

**Up-to-date** Hundreds of new publications on COVID-19 are released every day, and a dataset like CORD-19 can quickly become irrelevant without regular updates. CORD-19 has been updated daily since May 26. A processing pipeline that produces consistent results day to day is vital to maintaining a changing dataset. That is, the metadata and full text parsing results must be reproducible, identifiers must be persistent between releases, and changes or new features should ideally be compatible with previous versions of the dataset.

**Handles data from multiple sources** Papers from different sources must be integrated and harmonized. Each source has its own metadata format, which must be converted to the CORD-19 format, while addressing any missing or extraneous fields. The processing pipeline must also be flexible to adding new sources.

---

on the CORD-19 landing page.

**Clean canonical metadata** Because of the diversity of paper sources, duplication is unavoidable. Once paper metadata from each source is cleaned and organized into CORD-19 format, we apply the deduplication logic described in Section 2.2 to identify similar paper entries from different sources. We apply a conservative clustering algorithm, combining papers only when they have shared identifiers but no conflicts between any particular class of identifiers. We justify this because it is less harmful to retain a few duplicate papers than to remove a document that is potentially unique and useful.

**Machine readable full text** To provide accessible and canonical structured full text, we parse content from PDFs and associated paper documents. The full text is represented in S2ORC JSON format (Lo et al., 2020), a schema designed to preserve most relevant paper structures such as paragraph breaks, section headers, inline references, and citations. S2ORC JSON is simple to use for many NLP tasks, where character-level indices are often employed for annotation of relevant entities or spans. The text and annotation representations in S2ORC share similarities with BioC (Comeau et al., 2019), a JSON schema introduced by the BioCreative community for shareable annotations, with both formats leveraging the flexibility of character-based span annotations. However, S2ORC JSON also provides a schema for representing other components of a paper, such as its metadata fields, bibliography entries, and reference objects for figures, tables, and equations. We leverage this flexible and somewhat complete representation of S2ORC JSON for CORD-19. We recognize that converting between PDF or XML to JSON is lossy. However, the benefits of a standard structured format, and the ability to reuse and share annotations made on top of that format have been critical to the success of CORD-19.

**Observes copyright restrictions** Papers in CORD-19 and academic papers more broadly are made available under a variety of copyright licenses. These licenses can restrict or limit the abilities of organizations such as AI2 from redistributing their content freely. Although much of the COVID-19 literature has been made open access by publishers, the provisions on these open access licenses differ greatly across papers. Additionally, many open access licenses grant the ability to read, or "consume" the paper, but may be restrictive in

**Given a query:**

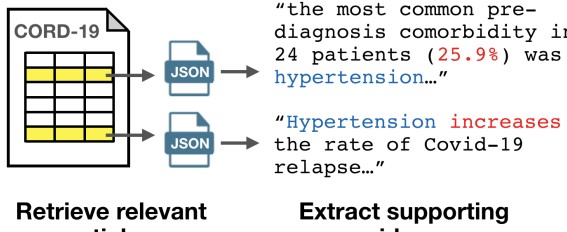

Figure 3: An example information retrieval and extraction system using CORD-19: Given an input query, the system identifies relevant papers (yellow highlighted rows) and extracts text snippets from the full text JSONs as supporting evidence.

other ways, for example, by not allowing replication of a paper or its redistribution for commercial purposes. The curator of a dataset like CORD-19 must pass on best-to-our-knowledge licensing information to the end user.

## 4 Research directions

We provide a survey of various ways researchers have made use of CORD-19. We organize these into four categories: *(i)* direct usage by clinicians and clinical researchers (§4.1), *(ii)* tools and systems to assist clinicians (§4.2), *(iii)* research to support further text mining and NLP research (§4.3), and *(iv)* shared tasks and competitions (§4.4).

### 4.1 Usage by clinical researchers

CORD-19 has been used by medical experts as a paper collection for conducting systematic reviews. These reviews address questions about COVID-19 include infection and mortality rates in different demographics (Han et al., 2020), symptoms of the disease (Parasa et al., 2020), identifying suitable drugs for repurposing (Sadegh et al., 2020), management policies (Yaacoub et al., 2020), and interactions with other diseases (Crisan-Dabija et al., 2020; Popa et al., 2020).

### 4.2 Tools for clinicians

Challenges for clinicians and clinical researchers during the current epidemic include *(i)* keeping up to to date with recent papers about COVID-19, *(ii)* identifying useful papers from historical coronavirus literature, *(iii)* extracting useful information from the literature, and *(iv)* synthesizing knowledge from the literature. To facilitate solutions to

these challenges, dozens of tools and systems over CORD-19 have already been developed. Most combine elements of text-based information retrieval and extraction, as illustrated in Figure 3. We have compiled a list of these efforts on the CORD-19 public GitHub repository[19] and highlight some systems in Table 2.[20]

### 4.3 Text mining and NLP research

The following is a summary of resources released by the NLP community on top of CORD-19 to support other research activities.

**Information extraction** To support extractive systems, NER and entity linking of biomedical entities can be useful. NER and linking can be performed using NLP toolkits like ScispaCy (Neumann et al., 2019) or language models like BioBERT-base (Lee et al., 2019) and SciBERT-base (Beltagy et al., 2019) finetuned on biomedical NER datasets. Wang et al. (2020) augments CORD-19 full text with entity mentions predicted from several techniques, including weak supervision using the NLM's Unified Medical Language System (UMLS) Metathesaurus (Bodenreider, 2004).

**Text classification** Some efforts focus on extracting sentences or passages of interest. For example, Liang and Xie (2020) uses BERT (Devlin et al., 2019) to extract sentences from CORD-19 that contain COVID-19-related radiological findings.

**Pretrained model weights** BioBERT and SciBERT have been popular pretrained LMs for COVID-19-related tasks. DeepSet has released a BERT-base model pretrained on CORD-19.[21] SPECTER (Cohan et al., 2020) paper embeddings computed using paper titles and abstracts are being released with each CORD-19 update. SeVeN relation embeddings (Espinosa-Anke and Schockaert, 2018) between word pairs have also been made available for CORD-19.[22]

**Knowledge graphs** The Covid Graph project[23] releases a COVID-19 knowledge graph built from mining several public data sources, including CORD-19, and is perhaps the largest current initiative in this space. Ahamed and Samad (2020) rely on entity co-occurrences in CORD-19 to construct a graph that enables centrality-based ranking of drugs, pathogens, and biomolecules.

### 4.4 Competitions and Shared Tasks

The adoption of CORD-19 and the proliferation of text mining and NLP systems built on top of the dataset are supported by several COVID-19-related competitions and shared tasks.

#### 4.4.1 Kaggle

Kaggle hosts the CORD-19 Research Challenge,[24] a text-mining challenge that tasks participants with extracting answers to key scientific questions about COVID-19 from the papers in the CORD-19 dataset. Round 1 was initiated with a set of open-ended questions, e.g., *What is known about transmission, incubation, and environmental stability?* and *What do we know about* COVID-19 *risk factors?*

More than 500 teams participated in Round 1 of the Kaggle competition. Feedback from medical experts during Round 1 identified that the most useful contributions took the form of article summary tables. Round 2 subsequently focused on this task of table completion, and resulted in 100 additional submissions. A unique tabular schema is defined for each question, and answers are collected from across different automated extractions. For example, extractions for risk factors should include disease severity and fatality metrics, while extractions for incubation should include time ranges. Sufficient knowledge of COVID-19 is necessary to define these schema, to understand which fields are important to include (and exclude), and also to perform error-checking and manual curation.

#### 4.4.2 TREC

The TREC-COVID[25] shared task (Roberts et al., 2020; Voorhees et al., 2020) assesses systems on their ability to rank papers in CORD-19 based on their relevance to COVID-19-related topics. Topics are sourced from MedlinePlus searches, Twitter conversations, library searches at OHSU, as well as from direct conversations with researchers, reflecting actual queries made by the community. To emulate real-world surge in publications and rapidly-

---

[19]https://github.com/allenai/cord19

[20]There are many Search and QA systems to survey. We have chosen to highlight the systems that were made publicly-available within a few weeks of the CORD-19 initial release.

[21]https://huggingface.co/deepset/covid_bert_base

[22]https://github.com/luisespinosaanke/cord-19-seven

[23]https://covidgraph.org/

[24]https://www.kaggle.com/allen-institute-for-ai/CORD-19-research-challenge

[25]https://ir.nist.gov/covidSubmit/index.html

| Task | Project | Link | Description |
|------|---------|------|-------------|
| **Search and discovery** | NEURAL COVIDEX | https://covidex.ai/ | Uses a T5-base (Raffel et al., 2019) unsupervised reranker on BM25 (Jones et al., 2000) |
| | COVIDSCHOLAR | https://covidscholar.org/ | Adapts Weston et al. (2019) system for entity-centric queries |
| | KDCOVID | http://kdcovid.nl/about.html | Uses BioSentVec (Chen et al., 2019) similarity to identify relevant sentences |
| | SPIKE-CORD | https://spike.covid-19.apps.allenai.org | Enables users to define "regular expression"-like queries to directly search over full text |
| **Question answering** | COVIDASK | https://covidask.korea.ac.kr/ | Adapts Seo et al. (2019) using BioASQ challenge (Task B) dataset (Tsatsaronis et al., 2015) |
| | AUEB | http://cslab241.cs.aueb.gr:5000/ | Adapts McDonald et al. (2018) using Tsatsaronis et al. (2015) |
| **Summariz-ation** | Vespa | https://cord19.vespa.ai/ | Generates summaries of paper abstracts using T5 (Raffel et al., 2019) |
| **Recommend-ation** | Vespa | https://cord19.vespa.ai/ | Recommends "similar papers" using Sentence-BERT (Reimers and Gurevych, 2019) and SPECTER embeddings (Cohan et al., 2020) |
| **Entailment** | COVID papers browser | https://github.com/gsarti/covid-papers-browser | Similar to KDCOVID, but uses embeddings from BERT models trained on NLI datasets |
| **Claim verification** | SciFact | https://scifact.apps.allenai.org | Uses RoBERTa-large (Liu et al., 2019) to find Support/Refute evidence for COVID-19 claims |
| **Assistive lit. review** | ASReview | https://github.com/asreview/asreview-covid19 | Active learning system with a CORD-19 plugin for identifying papers for literature reviews |
| **Augmented reading** | Sinequa | https://covidsearch.sinequa.com/app/covid-search/ | In-browser paper reader with entity highlighting on PDFs |
| **Visualization** | SciSight | https://scisight.apps.allenai.org | Network visualizations for browsing research groups working on COVID-19 |

Table 2: Publicly-available tools and systems for medical experts using CORD-19.

changing information needs, the shared task is organized in multiple rounds. Each round uses a specific version of CORD-19, has newly added topics, and gives participants one week to submit per-topic document rankings for judgment. Round 1 topics included more general questions such as *What is the origin of COVID-19?* and *What are the initial symptoms of COVID-19?* while Round 3 topics have become more focused, e.g., *What are the observed mutations in the SARS-CoV-2 genome?* and *What are the longer-term complications of those who recover from COVID-19?* Around 60 medical domain experts, including indexers from NLM and medical students from OHSU and UTHealth, are involved in providing gold rankings for evaluation. TREC-COVID opened using the April 1st CORD-19 version and received submissions from over 55 participating teams.

# 5 Discussion

Several hundred new papers on COVID-19 are now being published every day. Automated methods are needed to analyze and synthesize information over this large quantity of content. The computing community has risen to the occasion, but it is clear that there is a critical need for better infrastructure to incorporate human judgments in the loop. Extractions need expert vetting, and search engines and systems must be designed to serve users.

Successful engagement and usage of CORD-19 speaks to our ability to bridge computing and biomedical communities over a common, global cause. From early results of the Kaggle challenge, we have learned which formats are conducive to collaboration, and which questions are the most urgent to answer. However, there is significant work that remains for determining *(i)* which methods are best to assist textual discovery over the literature, *(ii)* how best to involve expert curators in the pipeline, and *(iii)* which extracted results convert to successful COVID-19 treatments and management policies. Shared tasks and challenges, as well as continued analysis and synthesis of feedback will hopefully provide answers to these outstanding questions.

Since the initial release of CORD-19, we have implemented several new features based on com-

munity feedback, such as the inclusion of unique identifiers for papers, table parses, more sources, and daily updates. Most substantial outlying features requests have been implemented or addressed at this time. We will continue to update the dataset with more sources of papers and newly published literature as resources permit.

## 5.1 Limitations

Though we aim to be comprehensive, CORD-19 does not cover many relevant scientific documents on COVID-19. We have restricted ourselves to research papers and preprints, and do not incorporate other types of documents, such as technical reports, white papers, informational publications by governmental bodies, and more. Including these documents is outside the current scope of CORD-19, but we encourage other groups to curate and publish such datasets.

Within the scope of scientific papers, CORD-19 is also incomplete, though we continue to prioritize the addition of new sources. This has motivated the creation of other corpora supporting COVID-19 NLP, such as LitCovid (Chen et al., 2020), which provide complementary materials to CORD-19 derived from PubMed. Though we have since added PubMed as a source of papers in CORD-19, there are other domains such as the social sciences that are not currently represented, and we hope to incorporate these works as part of future work.

We also note the shortage of foreign language papers in CORD-19, especially Chinese language papers produced during the early stages of the epidemic. These papers may be useful to many researchers, and we are working with collaborators to provide them as supplementary data. However, challenges in both sourcing and licensing these papers for re-publication are additional hurdles.

## 5.2 Call to action

Though the full text of many scientific papers are available to researchers through CORD-19, a number of challenges prevent easy application of NLP and text mining techniques to these papers. First, the primary distribution format of scientific papers – PDF – is not amenable to text processing. The PDF file format is designed to share electronic documents rendered faithfully for reading and printing, and mixes visual with semantic information. Significant effort is needed to coerce PDF into a format more amenable to text mining, such as JATS

XML,[26] BioC (Comeau et al., 2019), or S2ORC JSON (Lo et al., 2020), which is used in CORD-19. Though there is substantial work in this domain, we can still benefit from better PDF parsing tools for scientific documents. As a complement, scientific papers should also be made available in a structured format like JSON, XML, or HTML.

Second, there is a clear need for more scientific content to be made accessible to researchers. Some publishers have made COVID-19 papers openly available during this time, but both the duration and scope of these epidemic-specific licenses are unclear. Papers describing research in related areas (e.g., on other infectious diseases, or relevant biological pathways) have also not been made open access, and are therefore unavailable in CORD-19 or otherwise. Securing release rights for papers not yet in CORD-19 but relevant for COVID-19 research is a significant portion of future work, led by the PMC COVID-19 Initiative.[6]

Lastly, there is no standard format for representing paper metadata. Existing schemas like the JATS XML NISO standard[26] or library science standards like BIBFRAME[27] or Dublin Core[28] have been adopted to represent paper metadata. However, these standards can be too coarse-grained to capture all necessary paper metadata elements, or may lack a strict schema, causing representations to vary greatly across publishers who use them. To improve metadata coherence across sources, the community must define and agree upon an appropriate standard of representation.

## Summary

This project offers a paradigm of how the community can use machine learning to advance scientific research. By allowing computational access to the papers in CORD-19, we increase our ability to perform discovery over these texts. We hope the dataset and projects built on the dataset will serve as a template for future work in this area. We also believe there are substantial improvements that can be made in the ways we publish, share, and work with scientific papers. We offer a few suggestions that could dramatically increase community productivity, reduce redundant effort, and result in better discovery and understanding of the scientific literature.

---

[26]https://www.niso.org/publications/z3996-2019-jats
[27]https://www.loc.gov/bibframe/
[28]https://www.dublincore.org/specifications/dublin-core/dces/

Through CORD-19, we have learned the importance of bringing together different communities around the same scientific cause. It is clearer than ever that automated text analysis is not the solution, but rather one tool among many that can be directed to combat the COVID-19 epidemic. Crucially, the systems and tools we build must be designed to serve a use case, whether that's improving information retrieval for clinicians and medical professionals, summarizing the conclusions of the latest observational research or clinical trials, or converting these learnings to a format that is easily digestible by healthcare consumers.

## Acknowledgments

This work was supported in part by NSF Convergence Accelerator award 1936940, ONR grant N00014-18-1-2193, and the University of Washington WRF/Cable Professorship.

We thank The White House Office of Science and Technology Policy, the National Library of Medicine at the National Institutes of Health, Microsoft Research, Chan Zuckerberg Initiative, and Georgetown University's Center for Security and Emerging Technology for co-organizing the CORD-19 initiative. We thank Michael Kratsios, the Chief Technology Officer of the United States, and The White House Office of Science and Technology Policy for providing the initial seed set of questions for the Kaggle CORD-19 research challenge.

We thank Kaggle for coordinating the CORD-19 research challenge. In particular, we acknowledge Anthony Goldbloom for providing feedback on CORD-19 and for involving us in discussions around the Kaggle literature review tables project. We thank the National Institute of Standards and Technology (NIST), National Library of Medicine (NLM), Oregon Health and Science University (OHSU), and University of Texas Health Science Center at Houston (UTHealth) for co-organizing the TREC-COVID shared task. In particular, we thank our co-organizers – Steven Bedrick (OHSU), Aaron Cohen (OHSU), Dina Demner-Fushman (NLM), William Hersh (OHSU), Kirk Roberts (UTHealth), Ian Soboroff (NIST), and Ellen Voorhees (NIST) – for feedback on the design of CORD-19.

We acknowledge our partners at Elsevier and Springer Nature for providing additional full text coverage of papers included in the corpus.

We thank Bryan Newbold from the Internet Archive for providing feedback on data quality and helpful comments on early drafts of the manuscript.

We thank Rok Jun Lee, Hrishikesh Sathe, Dhaval Sonawane and Sudarshan Thitte from IBM Watson AI for their help in table parsing.

We also acknowledge and thank our collaborators from AI2: Paul Sayre and Sam Skjonsberg for providing front-end support for CORD-19 and TREC-COVID, Michael Schmitz for setting up the CORD-19 Discourse community forums, Adriana Dunn for creating webpage content and marketing, Linda Wagner for collecting community feedback, Jonathan Borchardt, Doug Downey, Tom Hope, Daniel King, and Gabriel Stanovsky for contributing supplemental data to the CORD-19 effort, Alex Schokking for his work on the Semantic Scholar COVID-19 Research Feed, Darrell Plessas for technical support, and Carissa Schoenick for help with public relations.

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

## A Table parsing results

There is high variance in the representation of tables across different paper PDFs. The goal of table parsing is to extract all tables from PDFs and represent them in HTML table format, along with associated titles and headings. In Table 3, we provide several example table parses, showing the high diversity of table representations across documents, the structure of resulting parses, and some common parse errors.

| PDF Representation | HTML Table Parse | Source & Description |
|---|---|---|

**Table 1**

| Effect | log-HR | SE×10 | P-value | HR | 95% CI |
|---|---|---|---|---|---|
| Female | 0 | | | 1 | |
| Male | 0.40 | 0.27 | < 0.001 | 1.50 | 1.40−1.60 |
| Age 65 | 0 | | | 1 | |
| Age − 65 | 0.09 | 0.01 | < 0.001 | 1.09 | 1.09−1.09 |
| covid-19 × Female | 0 | | | 1 | |
| covid-19 × Male | 0.18 | 0.73 | 0.05 | 1.20 | 1.00−1.44 |
| covid-19 × Age 65 | 0 | | | 1 | |
| covid-19 × Age − 65 | 0.04 | 0.03 | < 0.001 | 1.04 | 1.03−1.05 |

From Hothorn et al. (2020): Exact Structure; Minimal row rules

**Table 2**

| Time for surgery | Priority level | Functional urology surgeries in this category |
|---|---|---|
| 24 h | 1a, emergency | None |
| 72 h | 1b, urgent | Infected prosthesis/implant |
| 4 wk[a] | 2 | None |
| 3 mo[a] | 3 | None |
| >3 mo[a] | 4 | All the rest (Table 4) |

From López-Fando et al. (2020): Exact Structure; Colored rows

**Table 3**

| | SARS-CoV-2 serology test result | | | Relative risk (95% CI) | p value |
|---|---|---|---|---|---|
| | Positive | Negative | Indeterminate | | |
| Age group, years | | | | | |
| 5-9 (n=123) | 1 (0·8%) | 114 (92·7%) | 8 (6·5%) | 0·32 (0·11-0·63) | 0·0008 |
| 10-19 (n=332) | 32 (9·6%) | 295 (88·9%) | 5 (1·5%) | 0·86 (0·57-1·22) | 0·37 |
| 20-49 (n=1096) | 108 (9·9%) | 970 (88·5%) | 18 (1·6%) | 1 (ref) | .. |
| 50-64 (n=846) | 63 (7·4%) | 772 (91·3%) | 11 (1·3%) | 0·79 (0·57-1·04) | 0·090 |
| ≥65 (n=369) | 15 (4·1%) | 348 (94·3%) | 6 (1·6%) | 0·50 (0·28-0·78) | 0·0020 |
| Sex | | | | | |
| Female (n=1454) | 101 (6·9%) | 1333 (91·7%) | 20 (1·4%) | 1 (ref) | .. |
| Male (n=1312) | 118 (9·0%) | 1166 (88·9%) | 28 (2·1%) | 1·26 (1·00-1·58) | 0·054 |

From Stringhini et al. (2020): Minor span errors; Partially colored background with minimal row rules

**Table 4**

| | Number of study | | Prevalence (%) |
|---|---|---|---|
| Sex | | | |
| | male | 79 | 54.26 |
| | female | 79 | 45.82 |
| | Exposure history | 21 | 35.56 |
| | Signs and symptoms Fever | 66 | 79.84 |
| | Cough | 65 | 59.53 |
| | Fatigue or Myalgia | 56 | 33.46 |
| | Dyspnea | 56 | 31.48 |
| | Diarrhea | 52 | 10.71 |

From Fathi et al. (2020): Overmerge and span errors; Some section headers have row rules

**Table 5**

| Tests | Value | Reference Normal Range |
|---|---|---|
| SARS-CoV-2 PCR positive | 11 (33%) | |
| SARS-CoV-2 antibody positive | 27 (81%) | |
| SARs CoV-2 PCR and Antibody positive | 6 (18%) | |
| WBC in cells/uL, median (IQR) | 11,000 (8450, 14,400) | 4000-11,000 /uL |
| Hemoglobin in g/dL, median (IQR) | 11.3 (9.55, 12.5) | 10.5 - 14 g/dL |

From Kaushik et al. (2020): Over-splitting errors; Full row and column rules with large vertical spacing in cells

Table 3: A sample of table parses. Though most table structure is preserved accurately, the diversity of table representations results in some errors.