# OpenReview forum: "CORD-19: The COVID-19 Open Research Dataset"
_aclweb.org/ACL/2020/Workshop/NLP-COVID — NLP-COVID-2020_

### Official Review · AnonReviewer2 · 2020-07-04
**CORD-19 is an excellent resource with an impressive integration work for the research community to fight COVID-19.**

**Rating:** 9
**Confidence:** 5

**Review:**

The authors present the CORD-19 data set and describe how it has been developed and continues to be developed. The CORD-19 data set is a valuable resource that provides access to the latest literature about COVID-19 and coronaviruses and it is updated daily with over 200k downloads. The generation of the CORD-19 requires a coordinated integration and processing effort that is significant. The contribution of this corpus is of high significance and will have a strong impact on the biomedical domain and support the development, for instance, of COVID-19 vaccines. The manuscript is clearly written and it is easy to understand.

The effort in providing a version of the latest literature in formats that can be processed by text analytics methods is excellent, using the latest of the available technology to do so. In the paper, it is mentioned in the manuscript that there are some problems in turning tables into structured format and the authors provide examples of issues that they have found. Table processing is done by IBM, who has as well a method for table processing that seems to be resilient to the problems mentioned and would be relevant to consider it for table processing (https://arxiv.org/abs/1911.10683).

The authors give an example of conflict from which it can be inferred that the same DOI might be linked to two different PubMed identifiers, the reviewer is curious why this might be the case and if an example could be provided.

When you mention “Classification of CORD-19 papers to Microsoft Academic Graph”, is this classification done by a method provided by the authors? is this classification provided as meta-data?

During my review the only typo I could find is:
* “other research activity.” —> “other research activities.”?
* “by not allowing republication of **an** paper”, an —> a

Please consider the following guideline for NLM trademarks: https://www.nlm.nih.gov/about/trademarks.html

---

> ### Author Response · Authors · 2020-07-05
> **Thank you and replies to comments**
>
> Thank you for your review! We've addressed most comments in a revision. Detailed responses below.
>
> > The authors give an example of conflict from which it can be inferred that the same DOI might be linked to two different PubMed identifiers, the reviewer is curious why this might be the case and if an example could be provided.
>
> This happens for a couple hundred papers in the CORD-19 corpus. There can be two PubMed entries that correspond to the same DOI, sometimes due to duplicate entries in PubMed, but more often when a paper's abstract is available in two languages. In these cases, PubMed can have two entries for a paper with the same DOI. For example:
>
> https://pubmed.ncbi.nlm.nih.gov/32532586/
>
> https://pubmed.ncbi.nlm.nih.gov/32426303/
>
> Both of these papers have DOI: 10.1016/j.remn.2020.04.004; the first has an English abstract, the second Spanish. We keep these as separate entries in CORD-19.
>
> > When you mention “Classification of CORD-19 papers to Microsoft Academic Graph”, is this classification done by a method provided by the authors? is this classification provided as meta-data?
>
> This classification is performed in collaboration with the Microsoft Academic Graph research group. The methods used are described in Shen et al 2018. We have added the citation to the sentence to clarify. We provide a mapping to Microsoft Academic Graph identifiers as a supplementary file on the CORD-19 dataset landing page: https://www.semanticscholar.org/cord19/download.

---

### Official Review · AnonReviewer3 · 2020-07-04
**Overview of a highly important Covid-19 dataset**

**Rating:** 9
**Confidence:** 5

**Review:**

This is a paper that describes an important research dataset that has been produced during the Covid-19 epidemic. The CORD-19 collection is used for much research and some challenge evaluations. Even though this paper does not report any research results per se, and the paper is posted on the ArXiv preprint server, this version will give a citable description of the collection that will likely be widely referenced.

The authors describe well the process of dealing not only with the technical issues of processing heterogeneous scientific papers but also the non-technical issues, such as copyright and licensing.

The authors do not make any unreasonable claims, although I do question the value of this collection for non-computational researchers and clinicians. As the authors note, the collection is not complete, which is essential for clinical researchers and certainly for clinicians (who do not typically read primary research papers anyways, and tend to focus more on summations). But the dataset is of tremendous value to computational and informatics researchers, and that should be pronounced.

I appreciate the Discussion that points out the limitations of how scientific information is currently published, and how it could be improved. One other concern that could be addressed is how long the Allen Institute for AI, which is to be commended for this work, will continue to maintain this tremendously valuable resource.

---

> ### Author Response · Authors · 2020-07-05
> **Thank you and response**
>
> Thank you for your review!
>
> > I appreciate the Discussion that points out the limitations of how scientific information is currently published, and how it could be improved. One other concern that could be addressed is how long the Allen Institute for AI, which is to be commended for this work, will continue to maintain this tremendously valuable resource.
>
> We have automated the maintenance of the dataset, so will continue to update it for the foreseeable future. However, the answer to this question depends more on the value of the dataset to the community: whether the pandemic continues, whether we derive further value from mining these texts, whether the publishing community continues to support our efforts, and the costs / resources available for maintenance. We will try to balance all of these considerations.

---

### Official Review · AnonReviewer1 · 2020-07-05
**Excellent description of a critical COVID-19 dataset, some questions remaining**

**Rating:** 9
**Confidence:** 5

**Review:**

This manuscript describes an exemplary effort to address COVID-19 by bringing together much of the relevant literature into one corpus, CORD-19, and increasing its accessibility by providing a harmonized and standardized format convenient for use by automated tools. CORD-19 has been - and is likely to continue being - a critical resource for the scientific community to address COVID-19, and this manuscript not only reflects that importance, but also gives insight into the approach used, the design decisions taken, challenges encountered, use cases, shared tasks, and various discussion points. The manuscript is well-organized and readable, and (overall) an excellent case study in corpus creation. This manuscript is not only important for understanding the CORD-19 corpus and its enabling effect on current COVID-19 efforts, but is possibly also a historically important example of joint scientific efforts to address COVID-19.

Despite the critical importance of this dataset, there are several questions left unanswered by this manuscript, and it would be unfortunate to not address these before publication.

It would be useful to have a very clear statement of the purpose for CORD-19. The inclusion of SARS and MERS makes intuitive sense, but it is less clear why other coronaviruses that infect humans (e.g. HCoV-OC43) are not explicitly included - I am not a virologist, but neither will be most of the audience for this manuscript. While many of the articles that discuss these lesser known cornaviruses would be included anyway because they would also mention "coronavirus", this is not guaranteed.

While it seems appropriate for document inclusion to be query-based, it is important to consider the coverage of the query. The number of name variants in the literature for COVID-19 or SARS-CoV-2 is rather large, and not all of these documents will include other terms that will match, such as "coronavirus". For example, how would a document that mentions "SARS CoV-2" but none of the query terms listed be handled? This is not a theoretical case: the title and abstract for PMID 32584542 have this issue, and I was unable to locate this document in CORD-19. In addition to minor variations such as this, there are many examples of significant variations such as "HCoV-19", "nCoV-19" or even "COIVD". Are these cases worth considering? If not, can we quantify how much is lost? And if we can't quantify it, this is a limitation.

How is the following situation handled: querying source A returns a document (e.g. the source has full text and that matches), but the version in source B does not return it (e.g. the source only has title & abstract, and they do not match). From the description, I would assume that the version from source A is used and the version from source B is ignored; is any reasonably useful data lost by not explicitly querying source B for its version?

There are other efforts to provide a repository of scientific articles related to COVID-19, and it would be appropriate to mention these, if only to indicate why CORD-19 has unique value. I am aware of LitCovid (Chen Q, Allot A, Lu Z. Keep up with the latest coronavirus research. Nature. 2020;579(7798):193), are there others?

There are also non-COVID-19 efforts to provide a large percentage of the literature in formats appropriate for text mining or other processing. One is (Comeau, Donald C., et al. "PMC text mining subset in BioC: about three million full-text articles and growing." Bioinformatics 35.18 (2019): 3533-3535.), which not only provides the full text of a large percentage of the articles in PubMed Central, but it is also kept up-to-date and converts all documents into a straightforward standardized XML format appropriate for text mining. While this effort is single-source, it specifically addresses some of the issues encountered in the creation of CORD-19 and the representation aspect of the "Call to Action".

---

> ### Author Response · Authors · 2020-07-05
> **Thank you - new revision and responses**
>
> Thank you for the detailed review. We have uploaded a new revision which addresses most of the comments. More detailed explanations are provided below.
>
> > It would be useful to have a very clear statement of the purpose for CORD-19...
>
> > While it seems appropriate for document inclusion to be query-based, it is important to consider the coverage of the query...
>
> The reviewer brings up a good point. For any such text corpus, inclusion/exclusion is defined by the curators of the corpus, dramatically affecting how the corpus can be used. We adopted the initial query from Elsevier after consultation with other groups, including PMC. The query has evolved slightly since the beginning and may continue to evolve as needed. Although we don't explicitly include other coronaviruses in the search query, the current query should be catching most if not all of these papers (we match the search terms over title, abstract, *and* body text). PMC also performs some query expansion, which would increase coverage and also handle some of the lexical variations and misspellings (though not all) brought up by the reviewer.
>
> As for the other example brought up by the reviewer (PubMed 32584542); this paper is included in CORD-19 as of the latest versions (I checked 2020-07-04). The paper was published on 6/25, and is indexed in Medline. Papers outside of PMC take a longer period of time to make it into CORD-19 because of differences in the ingestion pipeline. PMC papers are queried directly from PMC every day, while papers from other sources are ingested by Semantic Scholar and have to go through our entire paper processing pipeline (which sometimes takes days to a week) before they become available in CORD-19. I hope this addresses some of the reviewer's concerns.
>
> > How is the following situation handled: querying source A returns a document (e.g. the source has full text and that matches), but the version in source B does not return it (e.g. the source only has title & abstract, and they do not match). From the description, I would assume that the version from source A is used and the version from source B is ignored; is any reasonably useful data lost by not explicitly querying source B for its version?
>
> In the case where source A and source B provide the same paper, and major metadata fields (DOIs and other identifiers) match, we cluster them together (preserving both A and B for provenance). The title and abstract are selected from the source that provides full text and the most unrestrictive license. If major metadata fields do not match, which sounds like the case described by the reviewer, we keep these as separate entries in CORD-19. We prefer underclustering to avoid the situation described by the reviewer (that of lost information).
>
> > There are other efforts to provide a repository of scientific articles related to COVID-19, and it would be appropriate to mention these, if only to indicate why CORD-19 has unique value. I am aware of LitCovid (Chen Q, Allot A, Lu Z. Keep up with the latest coronavirus research. Nature. 2020;579(7798):193), are there others?
>
> We have added a citation and description of LitCovid. We place additional emphasis on other areas of missing COVID-19 literature in the discussion (such as foreign language literature, works in the social sciences and other fields of study). Though we are not currently aware of any released corpora in these other areas, we would really appreciate if any of these are brought to our attention.
>
> > There are also non-COVID-19 efforts to provide a large percentage of the literature in formats appropriate for text mining or other processing. One is (Comeau, Donald C., et al. "PMC text mining subset in BioC: about three million full-text articles and growing." Bioinformatics 35.18 (2019): 3533-3535.), which not only provides the full text of a large percentage of the articles in PubMed Central, but it is also kept up-to-date and converts all documents into a straightforward standardized XML format appropriate for text mining. While this effort is single-source, it specifically addresses some of the issues encountered in the creation of CORD-19 and the representation aspect of the "Call to Action".
>
> BioC is very useful for text mining, and the full text representation in CORD-19 is very similar to BioC in some ways (character-level span annotations etc). We have added references and descriptions of BioC to both the "Design decisions" and the "Call to Action" sections of the paper.
>
> We adopt the JSON schema we use in CORD-19 from S2ORC (https://arxiv.org/abs/1911.02782), which is essentially an extended version of CORD-19 that covers all of the Semantic Scholar literature corpus. Some differences between CORD-19/S2ORC and BioC is the focus on full paper representation, with fields for bibliography entries, figure and table references, and so on included in S2ORC.

---

### Decision · Program_Chairs · 2020-07-04

**Decision:**

Accept

**Comment:**

This is a resource of significant importance, and a high-quality paper describing it.

We look forward to a presentation about this at the workshop later this week!

---

> ### Author Response · Authors · 2020-07-05
> **Looking forward to it!**
>
> Thank you! We look forward to being a part of the discussion.